# Accuracy of latest large language models in answering multiple choice questions in dentistry: A comparative study

**Huy Cong Nguyen[1], Hai Phong Dang[1], Thuy Linh Nguyen[1], Viet Hoang[2], Viet Anh Nguyen[1] ***

**1** Faculty of Dentistry, PHENIKAA University, Hanoi, Vietnam, **2** Faculty of Dentistry, Van Lang University, Ho Chi Minh City, Vietnam

* anh.nguyenviet1@phenikaa-uni.edu.vn

**Data Availability Statement:** All relevant data are within the manuscript and its Supporting Information files.

## Abstract

### Objectives

This study aims to evaluate the performance of the latest large language models (LLMs) in answering dental multiple choice questions (MCQs), including both text-based and image-based questions.

### Material and methods

A total of 1490 MCQs from two board review books for the United States National Board Dental Examination were selected. This study evaluated six of the latest LLMs as of August 2024, including ChatGPT 4.0 omni (OpenAI), Gemini Advanced 1.5 Pro (Google), Copilot Pro with GPT-4 Turbo (Microsoft), Claude 3.5 Sonnet (Anthropic), Mistral Large 2 (Mistral AI), and Llama 3.1 405b (Meta). $\chi^2$ tests were performed to determine whether there were significant differences in the percentages of correct answers among LLMs for both the total sample and each discipline (p < 0.05).

### Results

Significant differences were observed in the percentage of accurate answers among the six LLMs across text-based questions, image-based questions, and the total sample (p<0.001). For the total sample, Copilot (85.5%), Claude (84.0%), and ChatGPT (83.8%) demonstrated the highest accuracy, followed by Mistral (78.3%) and Gemini (77.1%), with Llama (72.4%) exhibiting the lowest.

### Conclusions

Newer versions of LLMs demonstrate superior performance in answering dental MCQs compared to earlier versions. Copilot, Claude, and ChatGPT achieved high accuracy on text-based questions and low accuracy on image-based questions. LLMs capable of handling image-based questions demonstrated superior performance compared to LLMs limited to text-based questions.

**Funding:** The author(s) received no specific funding for this work.

**Competing interests:** The authors have declared that no competing interests exist.

## Clinical relevance

Dental clinicians and students should prioritize the most up-to-date LLMs when supporting their learning, clinical practice, and research.

## 1. Introduction

Generative artificial intelligence (AI) large language models (LLM) have been widely applied in many fields of dentistry. Various applications include dental telemedicine, clinical decision support, administrative work, patient education, student education, scientific writing, and multilingual communication [1]. Additionally, generative AIs have been used to generate Synthesis Datasets for training robust AI models, which can be applied in dental research and education [2]. Furthermore, generative AI has been demonstrated to improve the performance of dental students in knowledge examinations compared to traditional literature research [3]. Beyond their use in answering clinical questions, LLMs are also being explored for their potential to generate questions that assess clinical reasoning skills, a crucial aspect of medical and dental education [4].

However, the accuracy of LLMs' responses to dental questions remains a concern and has been studied extensively. Studies report that the accuracy of LLMs in answering open-ended questions ranges from 52.5% to 71.7%, with responses occasionally being inaccurate, overly general, outdated, or lacking evidence-based support [5, 6]. For true or false questions, LLMs demonstrate lower accuracy compared to dentists, ranging from 57.3% to 78.0% [7, 8]. Regarding multiple-choice questions (MCQ), LLMs' accuracy varies from 42.5% to 80.7%, with ChatGPT 4.0 (OpenAI) proving the most accurate and Llama 2 (Meta) the least [9–11].

Studies have confirmed that within LLMs of the same developer, later versions consistently outperform older ones [9–12]. The field of generative AI is evolving rapidly, with new versions boasting increasingly powerful parameters being released in quick succession. However, previous studies on the accuracy of LLMs were performed on the older versions, which lacked the advanced multimodal capabilities now available. Furthermore, these studies excluded image-based questions because older LLM versions could not process image attachments in prompts [9, 11]. These critical limitations warrant further research exploring the full potential of current models.

Therefore, this study aims to evaluate the performance of the latest LLMs in answering dental MCQs, including both text-based and image-based questions. Model selection would be based on popularity, recency, multimodal capabilities, prominence in AI research and applications, accessibility, and ability to address domain-specific questions. The null hypothesis proposed that there is no difference in the accuracy of LLMs' answers to dental MCQs.

## 2. Materials and methods

### 2.1. Study design

This cross-sectional study adheres to the Strengthening the Reporting of Observational Studies in Epidemiology guideline (STROBE) and a research checklist for reporting AI studies [13, 14]. Ethical approval was not required for this study as it did not involve human participants. All 1490 MCQs available within two board review books for the United States (US) National Board Dental Examination were selected as the sampling base for the study [15, 16]. These books are widely recognized for their comprehensive coverage of the dental curriculum and

alignment with the national licensing exam. The US boasts numerous top-ranked dental schools, and its rigorous accreditation standards influence dental education globally [9]. Furthermore, these specific review books have an average rating of 4.7 out of 5 stars from 114 global ratings on Amazon, suggesting their acceptance as representative of global standards [17]. While the inclusion of all questions in the books captures their comprehensive nature, it also means that the number of questions varies across dental sub-disciplines, reflecting the emphasis placed on different topics within the books and potentially within the National Board Dental Examination itself. MCQs were chosen because they are easier to evaluate objectively compared to open-ended questions, and they reduce the likelihood of guessing compared to yes-or-no questions. Single-answer MCQs were chosen instead of multiple-answer ones to facilitate easier evaluation, especially in cases where the LLMs identify one correct answer among several.

This study evaluated six of the latest LLMs, including ChatGPT 4.0 omni (OpenAI) released in May 2024, Gemini Advanced 1.5 Pro (Google) released in May 2024, Copilot Pro with GPT-4 turbo (Microsoft) released in March 2024, Claude 3.5 Sonnet (Anthropic) released in June 2024, Mistral Large 2 (Mistral AI) released in July 2024, and Llama 3.1 405b (Meta) released in July 2024. Except for Llama 3.1 405b and Mistral Large 2, which were freely available but did not support image attachment to prompts, the remaining four LLMs required paid subscriptions and allowed image uploads along with prompts. The sample size was calculated based on a previous study by Chau et al., where ChatGPT 4.0 and ChatGPT 3.5 demonstrated accuracy rates of 0.807 and 0.683, respectively [9]. The calculation indicated that each LLM required a minimum of 319 questions to achieve a power of 0.95 and a significance level of 0.05.

## 2.2. Question input and accuracy evaluation

An independent assessor meticulously inputted the questions into the LLMs, maintaining the exact format, wording, and punctuation as presented in the source books between August 10, 2024, and August 22, 2024. This ensured consistency across all LLMs. After input, the same assessor collected the multiple-choice answers selected by ChatGPT. Each prompt was entered into a new conversation using a fresh LLM account and with previous conversations cleared to avoid bias from the LLM remembering past interactions. To minimize inaccuracies due to lengthy prompts, a maximum of 10 questions (or the 4000-character limit for Copilot) were included per prompt [11]. Another assessor independently marked the correct answers using the pre-existing answers from the book, serving as the benchmark for evaluating the LLMs' accuracy. In cases where the LLM was unable to provide an answer to a question, it would be considered an incorrect answer. LLMs not allowing image attachment to prompts would be deemed unable to answer image-based questions.

## 2.3. Statistical analysis

All data analyses were conducted using SPSS software (version 23.0; IBM, Armonk, NY). Frequencies and percentages of correct answers were calculated for each LLM across text-based, image-based questions, individual dental disciplines, and the total sample. $\chi^2$ tests were performed to determine whether there were significant differences in the percentages of correct answers among LLMs for both the total sample and each discipline, with a significance level of $\alpha = 0.05$. In cases where statistical significance was detected, a post-hoc analysis with Bonferroni correction was conducted to identify specific statistically significant differences between pairs of LLMs. While methods like logistic regression could potentially account for variations across question types and LLM capabilities, $\chi^2$ tests with Bonferroni corrections were deemed

**Table 1. Accuracy of large language models for text-based, image-based questions and the total sample.**

| Question type | ChatGPT | Gemini | Copilot | Claude | Mistral | Llama | p |
|---|---|---|---|---|---|---|---|
| Text-based | 1219 (84.5)[a,b] | 1134 (78.6)[c,d] | 1244 (86.2)[a] | 1223 (84.8)[a,b] | 1166 (80.8)[b,c] | 1079 (74.8)[d] | < 0.001* |
| Image-based | 29 (61.7)[a,b] | 15 (31.9)[b] | 30 (63.8)[a] | 29 (61.7)[a,b] | 0 (0.0)[c] | 0 (0.0)[c] | < 0.001* |
| Total | 1248 (83.8)[a] | 1149 (77.1)[b] | 1274 (85.5)[a] | 1252 (84.0)[a] | 1166 (78.3)[b] | 1079 (72.4)[c] | < 0.001* |

Results are presented as frequency (percentage).

* There were significant differences in the percentages of correct answers among large language models with p < 0.05.

Percentages with the same lowercase letter were not statistically different, as determined by the post-hoc analysis with Bonferroni correction.

more appropriate for this study due to their focus on comparing proportions and their robustness to small sample sizes in certain categories. Conversely, logistic regression focuses on predicting the likelihood of a correct answer based on multiple variables.

## 3. Results

Significant differences were observed in the percentage of accurate answers among the six LLMs across text-based questions, image-based questions, and the total sample (p<0.001). All 1443 text-based questions were answered by all six LLMs. For this question type, Copilot demonstrated the highest percentage of accurate answers (86.2%), while Llama had the lowest (74.8%). Post-hoc analyses revealed no significant differences in accuracy among Copilot, Claude, and ChatGPT; among Claude, ChatGPT, and Mistral; between Mistral and Gemini; and between Gemini and Llama (Table 1).

Out of 47 image-based questions, ChatGPT, Copilot, and Claude answered all, with Copilot demonstrating the highest accuracy (63.8%). Gemini only answered 19 questions that included diagrams or clinical photos, achieving an accuracy of 53.6%. However, it was unable to answer 28 questions with radiographs, generating an automated response instead (Fig 1). Post-hoc analyses showed no significant differences in accuracy for image-based questions among Copilot, Claude, and ChatGPT. For the total sample, Copilot, Claude, and ChatGPT demonstrated the highest accuracy, followed by Mistral and Gemini, with Llama exhibiting the lowest (Fig 2).

Comparing among disciplines, the accuracy of answers to general anatomic science questions was the highest (79.5%-93.5%), while answers to questions in dental anatomy &

image
! This image has been deleted.

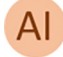 Sorry I can't help with that image. Try uploading another image or describing the image you tried to upload and I can help you that way.

**Fig 1. Gemini's automated response to image-based questions attaching radiographs.** AI, artificial intelligence. Note: This figure is a recreation for illustrative purposes only and is not an exact replica of the Gemini interface.

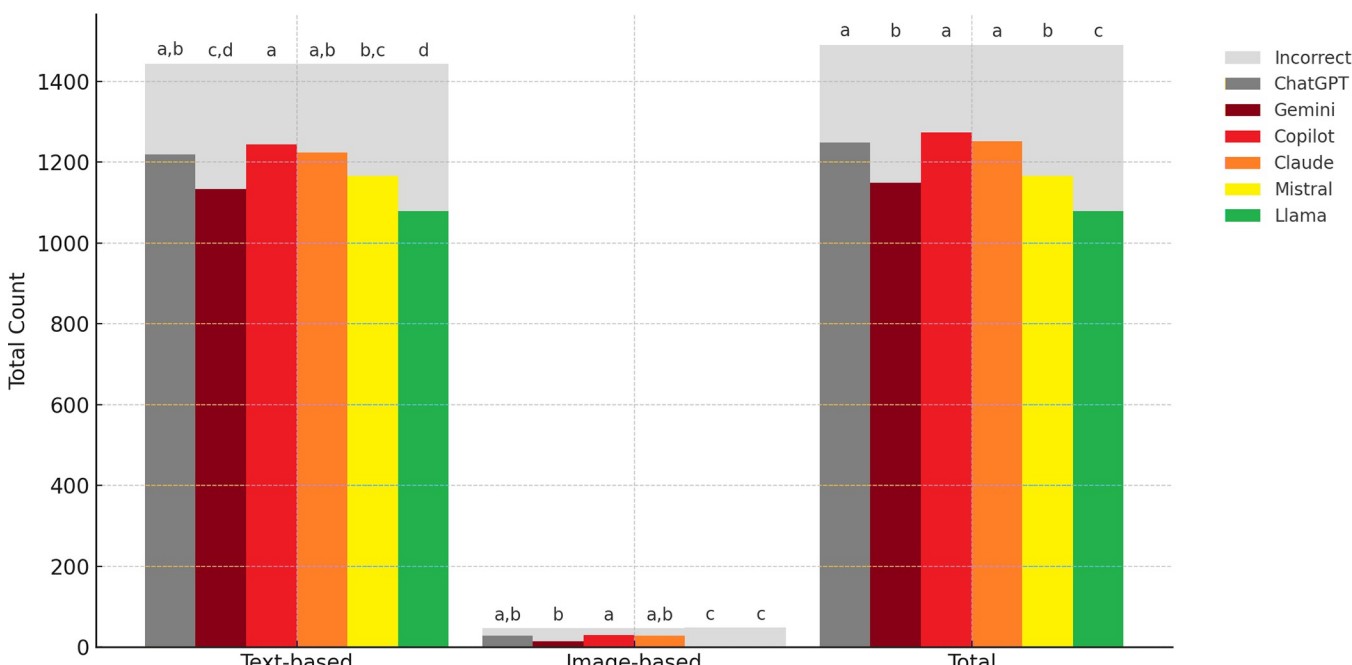

**Fig 2. Grouped stacked bar chart presenting the accuracy of large language models for text-based, image-based questions and the total sample.**
Percentages with the same lowercase letter were not statistically different, as determined by the post-hoc analysis with Bonferroni correction.

occlusion exhibited the lowest accuracy (49.5%-73.5%). Significant differences were observed in the accuracy among LLMs for answering questions of anatomic sciences (p < 0.001), biochemistry & physiology (p = 0.021), dental anatomy & occlusion (p < 0.001), oral & maxillofacial surgery (p = 0.043), orthodontics (p < 0.001), periodontics (p = 0.024) and pathology (p = 0.038) (Table 2 and Fig 3).

**Table 2. Accuracy of large language models for individual dental disciplines.**

| Discipline | ChatGPT | Gemini | Copilot | Claude | Mistral | Llama | p |
|---|---|---|---|---|---|---|---|
| Anatomic Sciences | 187 (93.5)[a] | 177 (88.5)[a,b] | 187 (93.5)[a] | 179 (89.5)[a,b] | 179 (89.5)[a,b] | 159 (79.5)[b] | < 0.001* |
| Biochemistry & Physiology | 182 (91.0)[a] | 164 (82.0)[a] | 182 (91.0)[a] | 180 (90.0)[a] | 182 (91.0)[a] | 173 (86.5)[a] | 0.021* |
| Microbiology & Pathology | 175 (87.5)[a] | 164 (82.0)[a] | 178 (89.0)[a] | 174 (87.0)[a] | 162 (81.0)[a] | 171 (85.5)[a] | 0.150 |
| Dental Anatomy & Occlusion | 136 (68.0)[a,b,c] | 112 (56.0)[c,d] | 147 (73.5)[b] | 140 (70.0)[a,b,c] | 114 (57.0)[a,c,d] | 99 (49.5)[d] | < 0.001* |
| Pharmacology | 51 (92.7)[a] | 48 (87.3)[a] | 49 (89.1)[a] | 50 (90.9)[a] | 49 (89.1)[a] | 48 (87.3)[a] | 0.934 |
| Operative Dentistry & Prosthodontics | 65 (68.4)[a] | 68 (71.6)[a] | 70 (73.7)[a] | 67 (70.5)[a] | 64 (67.4)[a] | 58 (61.1)[a] | 0.510 |
| Oral & Maxillofacial Surgery | 94 (83.9)[a] | 93 (83.0)[a] | 96 (85.7)[a] | 98 (87.5)[a] | 86 (76.8)[a] | 82 (73.2)[a] | 0.043* |
| Orthodontics | 122 (78.7)[a] | 119 (76.8)[a] | 130 (83.9)[a] | 130 (83.9)[a] | 112 (72.3)[a,b] | 90 (58.1)[b] | < 0.001* |
| Pediatric Dentistry | 25 (83.3)[a] | 20 (66.7)[a] | 25 (83.3)[a] | 25 (83.3)[a] | 24 (80.0)[a] | 20 (66.7)[a] | 0.322 |
| Endodontics | 42 (84.0)[a] | 37 (74.0)[a] | 42 (84.0)[a] | 43 (86.0)[a] | 41 (82.0)[a] | 40 (80.0)[a] | 0.692 |
| Periodontics | 28 (84.8)[a] | 26 (78.8)[a] | 30 (90.9)[a] | 29 (87.9)[a] | 28 (84.8)[a] | 20 (60.6)[a] | 0.024* |
| Radiology | 64 (85.3)[a] | 55 (73.3)[a] | 65 (86.7)[a] | 65 (86.7)[a] | 59 (78.7)[a] | 57 (76.0)[a] | 0.134 |
| Pathology | 45 (91.8)[a] | 37 (75.5)[a] | 44 (89.8)[a] | 43 (87.8)[a] | 40 (81.6)[a] | 35 (71.4)[a] | 0.038* |
| Patient Management & Public Health | 32 (88.9)[a] | 29 (80.6)[a] | 29 (80.6)[a] | 29 (80.6)[a] | 26 (72.2)[a] | 27 (75.0)[a] | 0.600 |

Results are presented as frequency (percentage).

* There were significant differences in the percentages of correct answers among large language models with p < 0.05.

Percentages with the same lowercase letter were not statistically different, as determined by the post-hoc analysis with Bonferroni correction.

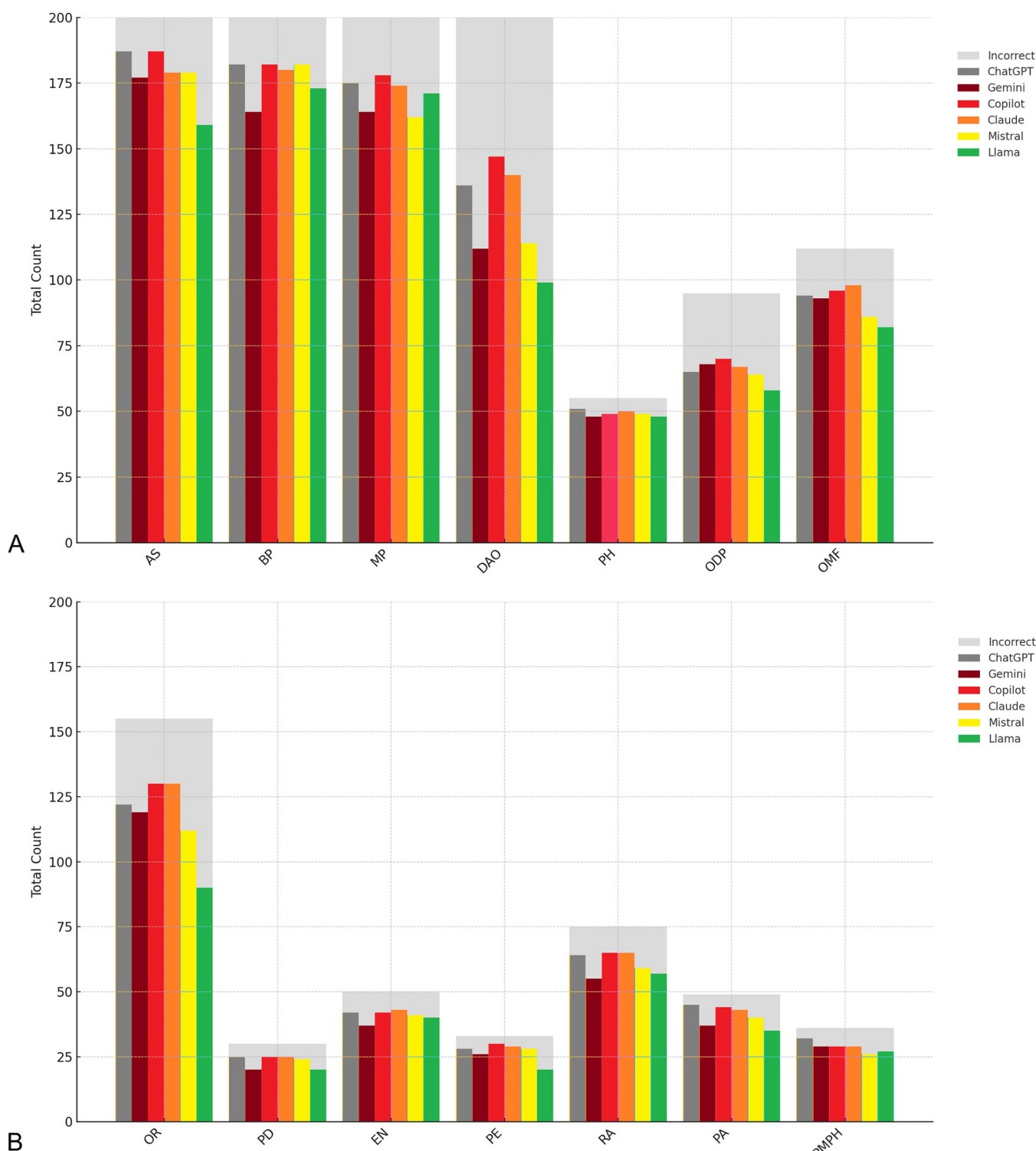

**Fig 3. Grouped stacked bar chart presenting the accuracy of large language models for individual dental disciplines.** (A) Anatomic Sciences (AS), Biochemistry & Physiology (BP), Microbiology & Pathology (MP), Dental Anatomy & Occlusion (DAO), Pharmacology (PH), Operative Dentistry & Prosthodontics (ODP), Oral & Maxillofacial Surgery (OMF). (B) Orthodontics (OR), Pediatric Dentistry (PD), Endodontics (EN), Periodontics (PE), Radiology (RA), Pathology (PA), Patient Management & Public Health (PMPH).

## 4. Discussion

Of the six evaluated LLMs, ChatGPT and Copilot share the same Generative Pre-trained Transformer architecture, while the remaining four utilize different architectures. This architectural similarity may explain the comparable accuracy observed between ChatGPT and Copilot across various question types and disciplines. However, it is also important to consider that variations in training data and fine-tuning strategies can still impact the performance of LLMs even when they share the same underlying architecture [18]. Interestingly, despite Copilot's fine-tuning and optimization in code generation [19], it performed slightly better in both dental text-based and image-based questions compared to ChatGPT. Variations in training data, such as the inclusion of domain-specific datasets or the emphasis on technical accuracy, may further contribute to Copilot's edge in handling specialized questions.

Priming, the practice of providing LLMs with initial, contextually relevant information, was not performed because the study of Fuch et al indicated the decreased effect of priming on advanced LLMs. Priming, the technique of providing LLMs with preliminary relevant information to the given context, was not utilized in this study due to its decreased effectiveness on advanced LLMs, as demonstrated by Fuch et al. [12].

The accuracy of ChatGPT in this study (83. 8%) was just slightly higher than in previous studies (76. 8%-80. 7%) [9–11]. This minor improvement might be attributed to the utilization of the same GPT-4 architecture, with potential advancements from GPT-4 to GPT-4 omni. However, Copilot's accuracy improved significantly from 72.6% in the previous study to 85.5% in this study [11]. This increase likely stems from the transition from GPT-3. 5 to GPT-4 turbo, aligning with the findings in other studies [9–11]. Noteworthy, Gemini and Llama showed a dramatic improvement in performance from 58.7% and 42.5% to 77.1% and 72.4%, respectively. This enhancement possibly results from the advancements from Gemini and Llama 2 7b to Gemini Advanced and Llama 3.1 405b. Besides the improved benchmarks of later versions, the enhanced training data may also contribute to the increase in accuracy.

The superior accuracy of Copilot in this study contrasts with previous findings where ChatGPT outperformed other LLMs in answering dental questions [8, 11]. However, the lowest accuracy observed for Llama is consistent with those prior studies. Additionally, Mistral's lower accuracy compared to ChatGPT and Claude aligns with the results of Benedict et al. [20], even though their study focused on musculoskeletal medicine questions.

The inherent complexity of image processing is a likely contributing factor to the lower accuracy observed in answers to image-based questions when compared to text-based questions. This study found lower accuracy (61. 7% to 63. 8%) for ChatGPT, Copilot, and Claude on image-based questions compared to reported accuracy (68% to 76. 7%) in other studies on image-based medical MCQs [21–23]. This lower accuracy may be attributed to limitations in LLM image interpretation algorithms, potentially stemming from their architectural design or insufficient exposure to diverse medical images during training, particularly those related to dentistry. Furthermore, the inherent ambiguity and subjectivity in interpreting certain images, even for human experts, could pose an additional challenge for these LLMs. Notably, the integration of image data combined with the use of ChatGPT-4V, a version specifically designed to process image inputs, led to a slight decrease in accuracy [21, 24]. This highlights the need to continue improving LLMs to be able to effectively integrate and utilize image data. Additionally, LLMs capable of processing all image-based questions also demonstrated superior performance in text-based questions compared to LLMs limited to processing only text-based questions.

The consistently high accuracy of LLMs in answering questions related to basic medical sciences, including anatomy, biochemistry & physiology, microbiology & pathology, and

pharmacology, aligns with the findings of Quad et al. [11]. This consistent performance can likely be attributed to the generalized knowledge base that these LLMs possess. The low accuracy of responses related to the dental anatomy & occlusion discipline may be due to the fact that this section includes many questions requiring spatial thinking, such as "How many cusp tips can be seen on the mandibular first molar when viewed directly from the buccal?". Additionally, this discrepancy may stem from gaps in the LLMs' training data, particularly regarding spatial reasoning and visualization skills necessary for questions related to dental anatomy and occlusion.

The high performance of the latest versions of Copilot, Claude, and ChatGPT in answering MCQs suggests that they may be a useful aid in dental education, and to a lesser extent, in supporting clinical practice and research. These models could be integrated into curricula as supplemental aids for self-directed learning, offering instant explanations and additional resources. However, to ensure the rigor of assessments is not compromised, educators should use LLMs strategically by focusing on enhancing students' critical thinking and problem-solving skills rather than relying solely on AI-generated responses. On the other hand, traditional assessment methods for evaluating core competencies in clinical decision-making should be maintained. Additionally, caution should be exercised and the use of these LLMs in dental examinations is strictly prohibited. This becomes even more important with the emergence of devices such as smartwatches or smart glasses, where the integration of powerful LLMs can open up entirely new possibilities for user interaction and assistance, increasing the risk of cheating in examinations [25–27]. To mitigate these risks, institutions can adopt strategies such as advanced proctoring technologies that monitor for unauthorized devices or altered examination formats that emphasize critical thinking and problem-solving over rote memorization.

The current study has several limitations. First, the number of image-based questions available within the selected resources was limited, lower than the calculated sample size for ideal comparisons. Despite this limitation, statistically significant differences were observed between certain LLMs on these questions. However, future research should focus on the ability to answer image-based questions, including radiographs and histopathology specimens, to enable more robust and generalizable comparisons of LLM performance in this critical domain. Second, the difficulty of the question bank was moderate to low, and the timeliness was low because the book was designed for the board exam. Future research should focus on more in-depth, more specialized, and more up-to-date questions. Third, the ability of LLMs to gather patient information and integrate multimodal data, such as combining text and images for comprehensive problem-solving, was not evaluated. Fourth, potential biases in the training data of LLMs may influence their performance on specialized topics. Furthermore, the study's reliance on US-based questions may limit generalizability. Future research should include diverse, non-US sources to assess the models' robustness across different educational contexts. Finally, clinical decision-making is much more complex than answering multiple-choice questions, requiring a combination of data including history taking, physical examination, and diagnostic tools. Clinicians must choose the most appropriate option from a list of options that they themselves provide, not those provided by the question. Therefore, high accuracy in answering multiple-choice questions should not be generalized to the performance of LLMs in clinical settings.

## 5. Conclusions

Newer versions of LLMs demonstrate superior performance in answering dental MCQs compared to earlier versions, with Copilot, Claude, and ChatGPT achieving high accuracy,

exceeding 80% on text-based questions. However, accuracy on image-based questions remains low, around 60%, underscoring the need for continuous updates to LLMs to better handle complex and specialized questions. LLMs capable of addressing both text and image-based queries outperformed those limited to text alone. Dental clinicians and students should prioritize using the most up-to-date LLMs while also balancing their reliance on these tools with the development of critical thinking skills to ensure sound decision-making in learning, clinical practice, and research.

## Supporting information

**S1 Dataset.**
(CSV)

## Author Contributions

**Conceptualization:** Viet Anh Nguyen.

**Data curation:** Huy Cong Nguyen, Hai Phong Dang, Viet Hoang, Viet Anh Nguyen.

**Formal analysis:** Huy Cong Nguyen, Hai Phong Dang, Viet Anh Nguyen.

**Investigation:** Huy Cong Nguyen, Viet Anh Nguyen.

**Methodology:** Viet Anh Nguyen.

**Software:** Huy Cong Nguyen, Hai Phong Dang.

**Supervision:** Thuy Linh Nguyen, Viet Anh Nguyen.

**Writing – original draft:** Viet Anh Nguyen.

**Writing – review & editing:** Viet Anh Nguyen.

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
