## [Decision Letter · Decision Letter 0]

1 Dec 2024

PONE-D-24-40356Accuracy of lastest large language models in answering multiple choice questions in dentistry: a comparative studyPLOS ONE

Dear Dr. Nguyen,

Thank you for submitting your manuscript to PLOS ONE. After careful consideration, we feel that it has merit but does not fully meet PLOS ONE’s publication criteria as it currently stands. Therefore, we invite you to submit a revised version of the manuscript that addresses the points raised during the review process.

Please submit your revised manuscript by Jan 15 2025 11:59PM. If you will need more time than this to complete your revisions, please reply to this message or contact the journal office at plosone@plos.org. Please include the following items when submitting your revised manuscript:A rebuttal letter that responds to each point raised by the academic editor and reviewer(s). You should upload this letter as a separate file labeled 'Response to Reviewers'.A marked-up copy of your manuscript that highlights changes made to the original version. You should upload this as a separate file labeled 'Revised Manuscript with Track Changes'.An unmarked version of your revised paper without tracked changes. You should upload this as a separate file labeled 'Manuscript'.

We look forward to receiving your revised manuscript.

Kind regards,

Jinran Wu, PhD

Academic Editor

PLOS ONE

Journal Requirements:

Reviewers' comments:

Reviewer's Responses to Questions

**Comments to the Author**

1. Is the manuscript technically sound, and do the data support the conclusions?

Reviewer #1: Yes

2. Has the statistical analysis been performed appropriately and rigorously? 

Reviewer #1: No

3. Have the authors made all data underlying the findings in their manuscript fully available?

Reviewer #1: Yes

4. Is the manuscript presented in an intelligible fashion and written in standard English?

Reviewer #1: Yes

5. Review Comments to the Author

Reviewer #1: This is an important and timely study, addressing the evolving role of large language models (LLMs) in dental education and their comparative performance. While the manuscript offers valuable insights, there are areas that require further clarification and expansion to enhance the study's rigor and relevance.

The introduction is clear and contextualizes the importance of LLMs in dental education. However, the rationale for selecting the six specific LLMs should be more detailed. Were they chosen based on popularity, accessibility, or specific capabilities in handling dental questions? This will strengthen the justification for their inclusion. The claim that newer versions perform better than older ones is supported by references, but it would benefit from a brief discussion of the limitations of previous studies (e.g., exclusion of image-based questions).

The study design is robust, adhering to STROBE guidelines, but a few points require elaboration. The selection of MCQs from U.S. board review books is reasonable; however, explain why these books are considered representative of global standards. Were efforts made to ensure that the questions are balanced across sub-disciplines and difficulty levels? The study acknowledges the limited number of image-based questions (47). Discuss whether this sample size is sufficient for meaningful comparisons and how future research might address this limitation. While the use of χ² tests and Bonferroni corrections is appropriate, clarify why these methods were chosen over others, such as logistic regression, which might better account for variations across question types and LLM capabilities.

The results are well-organized and supported by comprehensive tables. To improve clarity and impact, expand on the reasons for variation in LLM performance across disciplines (e.g., why dental anatomy and occlusion questions exhibit lower accuracy). Could this be linked to the spatial reasoning required or the training data limitations of LLMs? Discuss why ChatGPT, Claude, and Copilot had similarly low accuracies for image-based questions. Was this due to limitations in image interpretation algorithms, training data, or question complexity? Figure 1 illustrating Gemini's response to image-based questions is helpful, but additional visuals (e.g., a confusion matrix or graphical comparison of discipline-specific accuracies) would enhance understanding.

The discussion effectively interprets results but lacks depth in some areas. Provide a deeper analysis of why Copilot outperformed ChatGPT despite both being based on GPT architecture. This could include differences in fine-tuning, training data, or optimization for specific tasks. Highlight practical implications of the study for dental educators. For example, how can LLMs be integrated into curricula without compromising the rigor of assessment? The point about the potential misuse of LLMs in examinations is valid but could be expanded. Discuss mitigation strategies, such as proctoring technologies or modified examination formats. While the limitations section is clear, it could include the need to evaluate the models' ability to integrate multimodal data (e.g., combining text and images), potential biases in LLM training data that may influence performance on specialized topics, and a suggestion to incorporate questions from non-U.S. sources to assess generalizability.

The conclusion succinctly summarizes the findings but could emphasize the need for continuous updates to LLMs to handle complex and specialized questions effectively and the importance of balancing LLM reliance with critical thinking skills in dental education and practice.

Minor revisions include ensuring consistency in formatting tables and figures (e.g., uniform font sizes and styles). While the language is clear, minor grammatical edits are needed to improve readability (e.g., "Copilot's accuracy showed a significant improvement" could be rephrased for conciseness). Some references (e.g., those cited for image-based question accuracy) are compelling but could be updated to include the most recent studies in medical and dental education.

6. PLOS authors have the option to publish the peer review history of their article (what does this mean?). If published, this will include your full peer review and any attached files.

Reviewer #1: No

---

## [Author Response · Author response to Decision Letter 0]

12 Dec 2024

December 12th, 2024

Dear Editorial Board, PLOS ONE,

We have revised the manuscript thoroughly according to the comments of the reviewers. Any revisions made in our manuscript document were highlighted in red. Please help us review the manuscript again.

Sincerely,

Reviewer #1: This is an important and timely study, addressing the evolving role of large language models (LLMs) in dental education and their comparative performance. While the manuscript offers valuable insights, there are areas that require further clarification and expansion to enhance the study's rigor and relevance.

The introduction is clear and contextualizes the importance of LLMs in dental education. However, the rationale for selecting the six specific LLMs should be more detailed. Were they chosen based on popularity, accessibility, or specific capabilities in handling dental questions? This will strengthen the justification for their inclusion. The claim that newer versions perform better than older ones is supported by references, but it would benefit from a brief discussion of the limitations of previous studies (e.g., exclusion of image-based questions).

Thank you for your comment. We have added the criteria for selecting the six LLMs and previous studies' limitations to the introduction section:

However, previous studies on the accuracy of LLMs were performed on the older versions, which lacked the advanced multimodal capabilities now available. Furthermore, these studies excluded image-based questions because older LLM versions could not process image attachments in prompts [8,10]. These critical limitations warrant further research exploring the full potential of current models.

Model selection would be based on popularity, recency, multimodal capabilities, prominence in AI research and applications, accessibility, and ability to address domain-specific questions. 

The study design is robust, adhering to STROBE guidelines, but a few points require elaboration. The selection of MCQs from U.S. board review books is reasonable; however, explain why these books are considered representative of global standards. Were efforts made to ensure that the questions are balanced across sub-disciplines and difficulty levels? 

Thank you for your comment. We have added the explanation for the use of U.S. Board Review Books as global standards:

All 1490 MCQs available within two board review books for the United States (US) National Board Dental Examination were selected as the sampling base for the study [14,15]. These books are widely recognized for their comprehensive coverage of the dental curriculum and alignment with the national licensing exam. The US boasts numerous top-ranked dental schools, and its rigorous accreditation standards influence dental education globally [8]. Furthermore, these specific review books have an average rating of 4.7 out of 5 stars from 114 global ratings on Amazon, suggesting their acceptance as representative of global standards [16]. While the inclusion of all questions in the books captures their comprehensive nature, it also means that the number of questions varies across dental sub-disciplines, reflecting the emphasis placed on different topics within the books and potentially within the National Board Dental Examination itself.

The study acknowledges the limited number of image-based questions (47). Discuss whether this sample size is sufficient for meaningful comparisons and how future research might address this limitation. 

Thank you for your comment. We have added more discussion about the limited number of image-based questions:

First, the number of image-based questions available within the selected resources was limited, lower than the calculated sample size for ideal comparisons. Despite this limitation, statistically significant differences were observed between certain LLMs on these questions. However, future research should focus on the ability to answer image-based questions, including radiographs and histopathology specimens, to enable more robust and generalizable comparisons of LLM performance in this critical domain.

While the use of χ² tests and Bonferroni corrections is appropriate, clarify why these methods were chosen over others, such as logistic regression, which might better account for variations across question types and LLM capabilities.

Thank you for your comment. We have added the reason for choosing χ² tests and Bonferroni corrections:

While methods like logistic regression could potentially account for variations across question types and LLM capabilities, χ² tests with Bonferroni corrections were deemed more appropriate for this study due to their focus on comparing proportions and their robustness to small sample sizes in certain categories. Conversely, logistic regression focuses on predicting the likelihood of a correct answer based on multiple variables.

The results are well-organized and supported by comprehensive tables. To improve clarity and impact, expand on the reasons for variation in LLM performance across disciplines (e.g., why dental anatomy and occlusion questions exhibit lower accuracy). Could this be linked to the spatial reasoning required or the training data limitations of LLMs? Discuss why ChatGPT, Claude, and Copilot had similarly low accuracies for image-based questions. Was this due to limitations in image interpretation algorithms, training data, or question complexity? 

Thank you for your suggestion. We have added more discussion about the reason for variation in LLM performance:

Additionally, this discrepancy may stem from gaps in the LLMs' training data, particularly regarding spatial reasoning and visualization skills necessary for questions related to dental anatomy and occlusion.

This lower accuracy may be attributed to limitations in LLM image interpretation algorithms, potentially stemming from their architectural design or insufficient exposure to diverse medical images during training, particularly those related to dentistry. Furthermore, the inherent ambiguity and subjectivity in interpreting certain images, even for human experts, could pose an additional challenge for these LLMs. 

Figure 1 illustrating Gemini's response to image-based questions is helpful, but additional visuals (e.g., a confusion matrix or graphical comparison of discipline-specific accuracies) would enhance understanding.

Thank you for your suggestions. We have added Fig. 2 and Fig.3 for better visualization.

Fig. 2. Grouped stacked bar chart presenting the accuracy of large language models for text-based, image-based questions and the total sample. Percentages with the same lowercase letter were not statistically different, as determined by the post-hoc analysis with Bonferroni correction.

Fig. 3. Grouped stacked bar chart presenting the accuracy of large language models for individual dental disciplines. (A) Anatomic Sciences (AS), Biochemistry & Physiology (BP), Microbiology & Pathology (MP), Dental Anatomy & Occlusion (DAO), Pharmacology (PH), Operative Dentistry & Prosthodontics (ODP), Oral & Maxillofacial Surgery (OMF). (B) Orthodontics (OR), Pediatric Dentistry (PD), Endodontics (EN), Periodontics (PE), Radiology (RA), Pathology (PA), Patient Management & Public Health (PMPH).

The discussion effectively interprets results but lacks depth in some areas. Provide a deeper analysis of why Copilot outperformed ChatGPT despite both being based on GPT architecture. This could include differences in fine-tuning, training data, or optimization for specific tasks. 

Thank you for your comment. We have provided a deeper analysis according to your comment:

However, it is also important to consider that variations in training data and fine-tuning strategies can still impact the performance of LLMs even when they share the same underlying architecture [17]. Interestingly, despite Copilot’s fine-tuning and optimization in code generation [18], it performed slightly better in both dental text-based and image-based questions compared to ChatGPT. Variations in training data, such as the inclusion of domain-specific datasets or the emphasis on technical accuracy, may further contribute to Copilot’s edge in handling specialized questions.

Highlight practical implications of the study for dental educators. For example, how can LLMs be integrated into curricula without compromising the rigor of assessment? The point about the potential misuse of LLMs in examinations is valid but could be expanded. Discuss mitigation strategies, such as proctoring technologies or modified examination formats. 

Thank you for your suggestion. We have added the practical implications for the study according to your comments:

These models could be integrated into curricula as supplemental aids for self-directed learning, offering instant explanations and additional resources. However, to ensure the rigor of assessments is not compromised, educators should use LLMs strategically by focusing on enhancing students’ critical thinking and problem-solving skills rather than relying solely on AI-generated responses. On the other hand, traditional assessment methods for evaluating core competencies in clinical decision-making should be maintained. 

To mitigate these risks, institutions can adopt strategies such as advanced proctoring technologies that monitor for unauthorized devices or altered examination formats that emphasize critical thinking and problem-solving over rote memorization.

While the limitations section is clear, it could include the need to evaluate the models' ability to integrate multimodal data (e.g., combining text and images), potential biases in LLM training data that may influence performance on specialized topics, and a suggestion to incorporate questions from non-U.S. sources to assess generalizability.

Thank you for your comment. We have add more discussion on these limitations to the study limitation:

Third, the ability of LLMs to gather patient information and integrate multimodal data, such as combining text and images for comprehensive problem-solving, was not evaluated. Fourth, potential biases in the training data of LLMs may influence their performance on specialized topics. Furthermore, the study's reliance on US-based questions may limit generalizability. Future research should include diverse, non-US sources to assess the models' robustness across different educational contexts.

The conclusion succinctly summarizes the findings but could emphasize the need for continuous updates to LLMs to handle complex and specialized questions effectively and the importance of balancing LLM reliance with critical thinking skills in dental education and practice.

Thank you for your comment. We have modified the conclusions according to your comment:

However, accuracy on image-based questions remains low, around 60%, underscoring the need for continuous updates to LLMs to better handle complex and specialized questions. LLMs capable of addressing both text and image-based queries outperformed those limited to text alone. Dental clinicians and students should prioritize using the most up-to-date LLMs while also balancing their reliance on these tools with the development of critical thinking skills to ensure sound decision-making in learning, clinical practice, and research.

Minor revisions include ensuring consistency in formatting tables and figures (e.g., uniform font sizes and styles). 

Thank you for your comment. The table's large size required a smaller font to ensure readability within the document. We will ensure consistency in formatting, including uniform font sizes and styles, during the final typesetting process.

While the language is clear, minor grammatical edits are needed to improve readability (e.g., "Copilot's accuracy showed a significant improvement" could be rephrased for conciseness). 

Thank you for your comment. We have carefully reviewed the manuscript and made the necessary grammatical edits to enhance readability and clarity.

Copilot's accuracy improved significantly from 72.6% in the previous study to 85.5% in this study [10]. 

Some references (e.g., those cited for image-based question accuracy) are compelling but could be updated to include the most recent studies in medical and dental education.

Thank you for your comment. We have added more recent references to ensure the citations are comprehensive and reflect the latest advancements in the field.

---

## [Decision Letter · Decision Letter 1]

27 Dec 2024

PONE-D-24-40356R1Accuracy of lastest large language models in answering multiple choice questions in dentistry: a comparative studyPLOS ONE

Dear Dr. Nguyen,

Thank you for submitting your manuscript to PLOS ONE. After careful consideration, we feel that it has merit but does not fully meet PLOS ONE’s publication criteria as it currently stands. Therefore, we invite you to submit a revised version of the manuscript that addresses the points raised during the review process.

We look forward to receiving your revised manuscript.

Kind regards,

Jinran Wu, PhD

Academic Editor

PLOS ONE

Journal Requirements:

Reviewers' comments:

Reviewer's Responses to Questions

**Comments to the Author**

1. If the authors have adequately addressed your comments raised in a previous round of review and you feel that this manuscript is now acceptable for publication, you may indicate that here to bypass the “Comments to the Author” section, enter your conflict of interest statement in the “Confidential to Editor” section, and submit your "Accept" recommendation.

Reviewer #1: All comments have been addressed

Reviewer #2: (No Response)

2. Is the manuscript technically sound, and do the data support the conclusions?

Reviewer #1: Yes

Reviewer #2: Yes

3. Has the statistical analysis been performed appropriately and rigorously? 

Reviewer #1: N/A

Reviewer #2: Yes

4. Have the authors made all data underlying the findings in their manuscript fully available?

Reviewer #1: No

Reviewer #2: Yes

5. Is the manuscript presented in an intelligible fashion and written in standard English?

Reviewer #1: Yes

Reviewer #2: Yes

6. Review Comments to the Author

Reviewer #1: The authors have addressed all my concerns.

Congrats! Before the final acceptance, please polish the English writing.

Reviewer #2: The article aims to evaluate the accuracy of responses provided by current-generation Large Language Models (LLMs) to multiple-choice questions (MCQs) in the field of dentistry. The study is interesting both in terms of its chosen subject and methodological approach. Considering the lack of literature on the potential impact of LLMs in dental education and clinical practice, this work may provide a significant contribution. Methodologically, the performances of different LLMs have been comparatively and statistically analyzed. Overall, the article is well-structured, fluent, and written in a clear language.

I have only two suggestions.

First, the authors emphasize that they always used the most up-to-date LLM. However, given the rapid evolution of LLMs, it should be made clearer that this emphasis applies to August 2024. Currently, it is only mentioned once in the methodology section and not stated in the abstract.

The other suggestion concerns clinical reasoning mentioned within the article. There are studies in which LLMs are not only used to solve questions but also to produce questions utilizing clinical reasoning. At least a brief mention of this point could be included in the introduction regarding its relevance for medical/dental education. There are studies in the literature on LLM prompts designed to measure clinical reasoning skills. A recent study has even generated Script Concordance Test questions, for example:

Kıyak, Y. S., & Emekli, E. (2024). Using Large Language Models to Generate Script Concordance Test in Medical Education: ChatGPT and Claude. Spanish Journal of Medical Education, 6(1). https://doi.org/10.6018/edumed.636331

7. PLOS authors have the option to publish the peer review history of their article (what does this mean?). If published, this will include your full peer review and any attached files.

Reviewer #1: No

Reviewer #2: No

---

## [Author Response · Author response to Decision Letter 1]

27 Dec 2024

December 12th, 2024

Dear Editorial Board, PLOS ONE,

We have revised the manuscript thoroughly according to the comments of the reviewers. Any revisions made in our manuscript document were highlighted in red. Please help us review the manuscript again.

Sincerely,

Reviewer #1: The authors have addressed all my concerns.

Congrats! Before the final acceptance, please polish the English writing.

Thank you for your commendation. We have carefully reviewed the manuscript for grammar and language errors.

Reviewer #2: The article aims to evaluate the accuracy of responses provided by current-generation Large Language Models (LLMs) to multiple-choice questions (MCQs) in the field of dentistry. The study is interesting both in terms of its chosen subject and methodological approach. Considering the lack of literature on the potential impact of LLMs in dental education and clinical practice, this work may provide a significant contribution. Methodologically, the performances of different LLMs have been comparatively and statistically analyzed. Overall, the article is well-structured, fluent, and written in a clear language.

I have only two suggestions.

First, the authors emphasize that they always used the most up-to-date LLM. However, given the rapid evolution of LLMs, it should be made clearer that this emphasis applies to August 2024. Currently, it is only mentioned once in the methodology section and not stated in the abstract.

Thank you for your comment. We have added this clarification to the abstract:

This study evaluated six of the latest LLMs as of August 2024, including ChatGPT 4.0 omni (OpenAI), Gemini Advanced 1.5 Pro (Google), Copilot Pro with GPT-4 Turbo (Microsoft), Claude 3.5 Sonnet (Anthropic), Mistral Large 2 (Mistral AI), and Llama 3.1 405b (Meta). 

The other suggestion concerns clinical reasoning mentioned within the article. There are studies in which LLMs are not only used to solve questions but also to produce questions utilizing clinical reasoning. At least a brief mention of this point could be included in the introduction regarding its relevance for medical/dental education. There are studies in the literature on LLM prompts designed to measure clinical reasoning skills. A recent study has even generated Script Concordance Test questions, for example:

Kıyak, Y. S., & Emekli, E. (2024). Using Large Language Models to Generate Script Concordance Test in Medical Education: ChatGPT and Claude. Spanish Journal of Medical Education, 6(1). https://doi.org/10.6018/edumed.636331

Thank you for your suggestion. We have revised the introduction section and added your suggested reference:

Beyond their use in answering clinical questions, LLMs are also being explored for their potential to generate questions that assess clinical reasoning skills, a crucial aspect of medical and dental education [4].

---

## [Editor Report · Decision Letter 2]

30 Dec 2024

Accuracy of lastest large language models in answering multiple choice questions in dentistry: a comparative study

PONE-D-24-40356R2

Dear Dr. Nguyen,

We’re pleased to inform you that your manuscript has been judged scientifically suitable for publication and will be formally accepted for publication once it meets all outstanding technical requirements.

Kind regards,

Jinran Wu, PhD

Academic Editor

PLOS ONE

---

## [Editor Report · Acceptance letter]

17 Jan 2025

PONE-D-24-40356R2 

PLOS ONE

Dear Dr. Nguyen, 

I'm pleased to inform you that your manuscript has been deemed suitable for publication in PLOS ONE. Congratulations! Your manuscript is now being handed over to our production team.

Kind regards, 

on behalf of

Dr. Jinran Wu 

Academic Editor

PLOS ONE